# Cystic Fibrosis and Oxidative Stress: The Role of CFTR

**DOI:** 10.3390/molecules27165324

**Published:** 2022-08-21

**Authors:** Evelina Moliteo, Monica Sciacca, Antonino Palmeri, Maria Papale, Sara Manti, Giuseppe Fabio Parisi, Salvatore Leonardi

**Affiliations:** 1Pediatric Respiratory Unit, Department of Clinical and Experimental Medicine, San Marco Hospital, University of Catania, Viale Carlo Azeglio Ciampi sn, 95121 Catania, Italy; 2Pediatric Unit, Department of Human and Pediatric Pathology “Gaetano Barresi”, AOUP G. Martino, University of Messina, Via Consolare Valeria, 1, 98124 Messina, Italy

**Keywords:** cystic fibrosis, oxidative stress, cystic fibrosis transmembrane conductance regulator, antioxidant

## Abstract

There is substantial evidence in the literature that patients with cystic fibrosis (CF) have higher oxidative stress than patients with other diseases or healthy subjects. This results in an increase in reactive oxygen species (ROS) and in a deficit of antioxidant molecules and plays a fundamental role in the progression of chronic lung damage. Although it is known that recurrent infection–inflammation cycles in CF patients generate a highly oxidative environment, numerous clinical and preclinical studies suggest that the airways of a patient with CF present an inherently abnormal proinflammatory milieu due to elevated oxidative stress and abnormal lipid metabolism even before they become infected. This could be directly related to cystic fibrosis transmembrane conductance regulator (CFTR) deficiency, which appears to produce a redox imbalance in epithelial cells and extracellular fluids. This review aims to summarize the main mechanism by which CFTR deficiency is intrinsically responsible for the proinflammatory environment that characterizes the lung of a patient with CF.

## 1. Introduction

Cystic fibrosis (CF) is still today the most common lethal genetic disease with autosomal recessive inheritance in the Caucasian population, with a prevalence of 1 case per 2500 live births [1]. The disease is caused by a mutation in the cystic fibrosis transmembrane conductance regulator (CFTR) gene that causes the CFTR protein to become dysfunctional. When the protein is not working correctly, there is reduced transport of chloride ions with consequent dysregulation of epithelial lining fluid (mucus) transport in the lung, pancreas and other organs [2]. 

There are more than 2000 different mutations in the gene encoding the CFTR protein [3]. Among these, seven main groups have been identified based on the type of DNA alteration that characterizes the mutation. Class I mutations induce a block of protein synthesis. Class II mutations, of which the more common F508del mutation is part, synthesize a misfolded CFTR protein, leading to the failure of maturation and trafficking to the cell surface. Mutations of class III, also termed as “gating defect”, affect the activation of ion transport function, while mutations of class IV reduce the number of chloride ions transported through pore channels. Class V mutations allow the synthesis of the protein in reduced quantities. Class VI mutations produce unstable CFTR with a short half-life. Class VII mutations, recently introduced, interfere with mRNA splicing, leading to the absence of full-length mature RNA, so the CFTR protein is totally absent, as occurs in class I mutations [4,5,6,7].

The presence of so many mutations reflects the extremely variable phenotypes: some subjects have a severe clinical presentation, and their life expectancy is dependent on lung or liver transplantation. On the other extreme, there are patients who manifest the pathology later in life, or early but with mild symptoms or even without any [8].

Among the different organs affected, in the lungs, the accumulation of thick mucus decreases the ciliary mucus clearance function, favoring colonization by numerous germs, first bacteria, leading to infection, inflammation and other complications. The inflammation is a self-amplifying process: a vicious circle is established and constitutes a chronic challenge to the integrity of airway epithelial cells [9]. Oxidative stress is a key element contributing to persistent cellular damage and preventing proper airway remodeling.

Oxidative stress is a complex process in which excess reactive oxygen species (ROS) affect, either directly or indirectly, all structural and functional components of cells at a molecular level [10,11,12]. This arises because the production of these chemical species is increased and/or because the physiological defense capacity towards them, thanks to the antioxidant system, is reduced. Changes in the balance between oxidant and antioxidant substances are considered a normal part of cell physiology; many cellular signaling pathways, in fact, are regulated by changes in redox balance [13]. In CF patients, malabsorption of dietary antioxidants, induced by exocrine pancreatic insufficiency and by a decrease in bile acids, and the inability of cells with the CFTR mutation to efflux glutathione (GSH) play an essential role in the systemic redox imbalance already exacerbated by the excessive release of oxidants by neutrophils [14]. This sustained redox imbalance leads to the establishment of an oxidizing environment that causes the oxidation of proteins, DNA, lipids and other metabolites with the consequent alteration of various signaling pathways [13]. 

In consideration of the pathogenetic mechanism described above, it is logical that the optimization of the antioxidant and anti-inflammatory status represents an important goal in patients with CF. There is substantial evidence that antioxidant supplementation positively influences the outcome of CF patients, especially in terms of a reduction in pulmonary exacerbation, but the efficacy is limited and transient [15,16]. New therapeutic strategies are therefore necessary and are under study. 

CFTR-targeted therapeutics, mainly responsible for the increase in life expectancy that has occurred in recent years, in association with antioxidant and anti-inflammatory therapies, appear to be the only weapon to reduce the underlying inflammatory state that leads to progressive lung damage.

Ivacaftor was the first drug able to act on the causes of the disease by improving the function of the defective protein. It is suitable for gating mutations (class III) in the CFTR gene. Lumacaftor/ivacaftor was the first drug used for the defect in the processing and transport of the CFTR protein in patients with CF with a double copy of the F508del mutation. Last but not least, we can mention the triple combination elaxacaftor/tezacaftor/ivacaftor, which works as a modulator of the CFTR protein that is defective and therefore responsible for the symptoms of the disease [17,18,19].

This review aims to summarize the main mechanism by which CFTR deficiency is intrinsically responsible for the proinflammatory environment that characterizes the lung of a patient with CF.

A combination of antioxidant, anti-inflammatory and CFTR-targeted therapeutics could be required for full correction of the CF phenotype to decrease the basic inflammatory status, improving the disease outcome.

## 2. Literature Search Methodology

Literature searches for specific research were conducted using the PubMed database with keywords such as “cystic fibrosis”, “oxidative stress”, “cystic fibrosis transmembrane conductance regulator” and “antioxidant”. We included review articles, meta-analyses, case–control studies, case reports and letters to the editor, including only papers over the last 10 years and published in English. Only studies specifically correlating oxidative stress and cystic fibrosis were considered. The review was completed by searching for bibliographic references and definitions of the topic described above.

## 3. Evidence from Literature

One of the main determinants of progressive lung damage in CF is represented by chronic oxidative stress, which leads to the establishment of an intrinsically proinflammatory environment. The mechanisms behind this are still partially unknown, but several studies have shown the direct implication of CFTR protein dysfunction, mainly in the lungs, but also in extrapulmonary tissues such as the pancreas and intestine [20].

Several molecular mechanisms have been proposed to explain the link between CFTR deficiency and oxidative stress (Figure 1). 

There is some evidence that the efflux of GSH out of cells is a chloride-dependent mechanism involving the CFTR channel. Indeed, CFTR shares a structural similarity with ABCC proteins, which normally export glutathione and/or glutathione S-conjugates [13]. Glutathione (GSH) is a tripeptide with antioxidant properties consisting of cysteine, glycine and glutamic acid. It represents the first-line defense of the lung against oxidative stress-induced damage, and its availability inside the cell is fundamental to sustaining a good redox state. The ratio between reduced and oxidized glutathione is an indicator of the cellular redox state and describes the antioxidative capacity of cells [21].

Unsurprisingly, in patients with CF, low CFTR activity is correlated with GSH deficiency, resulting in an altered extracellular ratio between oxidized and reduced glutathione [22,23,24]; oxidized glutathione species are significantly elevated, and there is an inadequate response to neutrophil-mediated oxidative stress during infections. Rather, the reactive oxidant species produced by neutrophils, including myeloperoxidase (MPO)-derived hypochlorous acid, contribute to the oxidation of glutathione, leading to a vicious cycle. Dickerhof N et al. demonstrated that the pharmacological inhibition of MPO by orally administrated AZM198 decreases oxidative stress and improves infection outcomes in mice with CF-like inflammation without interfering with the clearance of bacteria [25]. Still, few studies have been conducted or are ongoing on the beneficial effects of direct GSH supplementation in CF. For example, Calabrese et al. studied the possible beneficial effects of long-term treatment with inhaled glutathione [26]. Hewson et al. established that exogenous administration of γ-glutamylcysteine (GGC), the immediate precursor of glutathione, can increase intracellular levels of total glutathione and protect CF cells from lipopolysaccharide (LPS)-induced cell damage [27].

It has also been demonstrated that the administration of N-acetylcysteine (NAC), the acetylated form of the amino acid L-cysteine and a precursor to glutathione, is able to reduce the redox imbalance, increasing the GSH level. Furthermore, an influence of NAC on nuclear factor kappa-light-chain-enhancer of activated B cells (NFkB) activation was observed [14].

Mutated CFTR is associated with the alteration of some signal transduction pathways at a cellular level, such as that of NFkB, required for the transcription of various proinflammatory molecules. NFkB overexpression is an intrinsic underlying feature of the patient with cystic fibrosis and is exacerbated by hyperproduction of ROS and by bacterial stimulation on the cell surface that induces further activation. Moreover, the CFTR mutation is also associated with reduced production of peroxisome proliferator-activated receptor (PPAR), a transcription factor that normally counteracts the action of NFKB [14,20]. This results in increased production of oxidizing molecules and proinflammatory cytokines such as IL-1, TNF, IL-6 and IL-17A. 

In normal cell physiology, under conditions of increased oxidant production, a series of pathways are activated that play an active role in the suppression of inflammatory signaling. Among these, the most important is the Nuclear factor erythroid 2-related factor 2 (Nrf2) pathway, an antagonist of proinflammatory transcription factors such as NFkB. Following the hyperactivation of the inflammatory response, the Kelch-like-ECH-associated protein 1 (KEAP1) protein, which normally binds Nrf2 in the cytoplasm of cells, oxidizes and dissociates from Nrf2, allowing its subsequent transcriptional activation, which leads to the production of over 200 antioxidant and detoxifying proteins; these include heme oxygenase-1 (HO-1), NAD(P)H quinone oxidoreductase 1 (NQO1), glutamate–cysteine ligase (GCL) and glutathione S transferase (GST). Nrf2-mediated HO-1 expression is also regulated by transcription factor BTB (TF BTB) and CNC Homology 1 (Bach1), which suppress HO-1 expression [26,28]. The heme oxygenase-1/carbon monoxide (HO-1/CO) pathway is essential to ensure a controlled immune response and effective bactericidal activity by monocytes and macrophages. The blunt activation of this pathway in CF patients therefore contributes to hyperinflammation and defective host defense against bacteria. Recent studies have shown that the administration of controlled doses of CO can induce HO1 by reducing lung hyperinflammation and oxidative stress [29]. Furthermore, CO stimulates autophagy [29], the cellular mechanism that is fundamental to efficient bacterial clearance by immune cells. Recent works prove that CFTR deficiency in macrophages and neutrophils results in an inability to kill bacteria and, thus, in limited autophagy activity [30].

There is much evidence that the Nrf2 pathway is dysfunctional in cells with mutated CFTR [13,28,29].

Laselva et al. demonstrated that the administration of dimethyl fumarate (DMF), an activator of the Nrf2 pathway, drastically reduced both the basal and stimulated expression of proinflammatory cytokines while also exerting an antioxidant effect [30].

Borcherding et al. demonstrated that the CFTR modulators VX-809 (Lumacaftor) and VX-661 (Tezacaftor) significantly increase Nrf2 activity in CF patients [31]; this could represent one of the mechanisms through which CFTR modulators mitigate the inflammatory response and oxidative stress.

Pellullo et al., in their study, used mRNA extracted from nasal epithelial cells to analyze the expression levels of the genes involved in oxidative stress. They found that the expression of Nrf2 mRNA and its targets, such as HO-1 and miR-125b, is upregulated in the nasal epithelia cells of CF patients compared to healthy subjects. This suggests that the protective mechanisms against oxidative stress may be functional but not sufficient to counteract the hyperproduction of ROS and the oxidative stress that characterizes the pathology. Moreover, the authors found that elevated HO-1 and miR-125b levels are associated with an improved FEV1 value, so they could be considered potential predictive biomarkers of CF clinical outcomes. The wide expression range of these markers could partly explain the phenotypic variability of CF, beyond the mutations of the CFTR gene itself [28].

Another mechanism that links CFTR deficiency and oxidative stress and that contributes to CF airways’ chronic damage is the alteration of lipid metabolism [29]. Thanks to several studies, it has been found that CF airway pathology is related to alterations in fatty acids, ceramides and cholesterol, but their role in the etiopathogenesis of CF pulmonary pathology is unclear.

An increased ratio of long-chain to very long chain ceramide species (LCC/VLCC), abnormalities in sphingosine phosphate (S1P) metabolism and, consequently, abnormal lipid levels in the blood and lungs are hallmarks of CF. Lipid synthesis is increased, whereas their catabolism is reduced, contributing to inflammation, oxidative stress and impaired autophagy. In this view, dyslipidemia should be considered a contributor to CF airways’ chronic damage, and thus, lipid metabolism should become an important therapeutic target [32,33,34].

Signorelli et al. have shown that modulating the synthesis of sphingolipids and hindering the accumulation of ceramide with Myrocin (Myr), a sphingolipid synthesis inhibitor, significantly reduces the accumulation of lipids, promotes the oxidation of fatty acids and reduces inflammation and oxidative stress, and it restores the defensive response against pathogen infection, which is defective in CF [35].

Veltman et al. examined the consequences of CFTR deficiency on lipid metabolism, highlighting how the alterations concerning the metabolism of fatty acids and ceramide induce a state of chronic oxidative stress, but also, in turn, chronic oxidative stress can cause a great imbalance in the metabolism of lipids. The new CFTR-modulating drugs considerably reduce the alterations of lipid balance, confirming the role of CFTR as a regulator of cellular lipid balance [36].

## 4. Discussion

The role of oxidative stress in the progression of lung injury in CF patients has been widely recognized and very well described in the literature (Table 1). It has been shown that in patients with CF, there is an important deficit of antioxidant molecules and an increase in oxidative stress [24,37]. The sustained imbalance between oxidant and antioxidant species induces chronic inflammation, which is the key element contributing to persistent cellular damage and preventing proper airway remodeling. Both hereditary and acquired factors, such as CFTR deficiency and persistent infections, contribute to abnormal and self-sustaining lung inflammation in CF. 

The central role of CFTR deficiency has been increasingly recognized in recent years. Mutations in the CFTR gene appear to make CF epithelial cells more susceptible to inflammation compared with healthy cells; for this reason, once an infection is introduced, it triggers the onset of mucosal damage and chronic airway infection. CFTR dysfunction, in fact, not only alters ion exchange and fluid secretion in the lungs but also causes the dysregulation of several signaling pathways, generating an innate oxidative state that, over time, could promote the loss of lung epithelial cell integrity. Impaired extracellular glutathione transport, alterations in lipid metabolism, dysregulation of the main pro- and anti-inflammatory signaling pathways and unbalanced autophagy are the main molecular mechanisms correlated with CFTR dysfunction in CF. Starting from this consolidated knowledge, numerous research groups have identified targets and strategies aimed at reducing the exaggerated immune response that causes chronic inflammation in CF, without altering the natural defenses against infection. Currently used drugs, such as steroidal and non-steroidal anti-inflammatories, mucolytics and antibiotics, reduce inflammation, improving the natural history of the disease; however, there are a lot of concerns about their chronic use because of their immunosuppressive effects that compromise the host’s defenses [29]. 

New therapeutic approaches are therefore needed and are currently being evaluated, with the aim of reducing the proinflammatory response in CF, preserving the host defense against microorganisms. The most promising results come from the use of CFTR modulators, which, in the last several years, have radically changed the natural history of CF. In our review, we found that the excellent results from the use of these drugs are related not only to the restoration of physiological ion exchanges, which improve mucociliary clearance, but also, given the close correlation between CFTR deficiency and oxidative stress, to the reduction in the basic inflammatory status that characterizes CF patients. Further studies are needed to confirm, through the evaluation of specific markers, whether CF patients treated with modulators have a significant reduction in oxidative stress. However, these therapies are targeted for specific mutations characterizing CF, and it is not known how long they will have this crucial role in containing the inflammatory response. Future therapeutic perspectives should include the use of additional antioxidant and anti-inflammatory drugs, in combination with CFTR modulators, which specifically target the altered signaling pathway, in order to obtain a more selective response without altering the local tissue defenses. 

## 5. Conclusions

Mutations in the CFTR gene produce an inherently proinflammatory cellular environment, in addition to repeated infections that set the stage for chronic airway infection and progressive loss of lung function. Although the molecular mechanisms underlying it are not fully understood, this aspect represents a central feature of the disease and, consequently, an important therapeutic target.

## Figures and Tables

**Figure 1 molecules-27-05324-f001:**
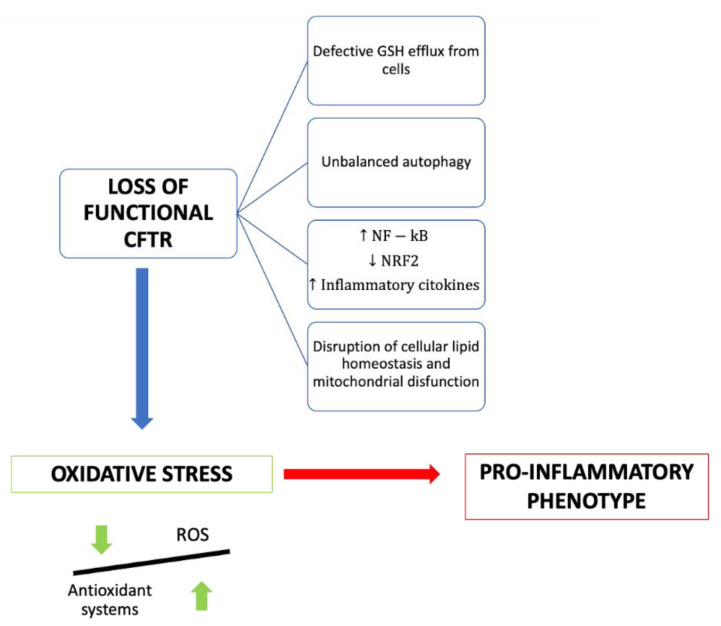
Summary of the consequences of the loss of functional CFTR in cystic fibrosis patients.

**Table 1 molecules-27-05324-t001:** Summary of the main studies pointing towards the involvement of oxidative stress in CF disease.

Authors	Type of Study	Aim of the Study	Materials and Methods	Main Findings
Checa et al. 2021 [10]	Research article	To identify oxidative stress modulators in CF airway epithelial cells	Unbiased genome-wide RNAi screen using a randomized siRNA library	The usefulness of combining unbiased genome-wide knockdown to uncover new genes/pathways involved in oxidative stress to identify and characterize new drugs.
Guerini et al. 2022 [14]	Review	To show the potential role of N-acetylcysteine in preventing and eliminating biofilms as an anti-inflammatory and antioxidant drug.	NA	It is possible to establish that this molecule offers great hope for the treatment of this disease.
Ciofu et al. 2014 [15]	Cochrane systematic review	To synthesize existing data on the effect of antioxidants such as vitamin C, vitamin E, ß-carotene, selenium and glutathione in CF disease.	Randomized controlled studies and quasi-randomized controlled studies of people with cystic fibrosis comparing antioxidants to placebo or standard care.	Intensive antibiotic treatment and other drugs used in CF patients make it very difficult to evaluate the usefulness of antioxidant therapy. Based on the available evidence, glutathione (administered either orally or by inhalation) appears to improve lung function.
Sagel et al. 2018 [16]	Research article	To evaluate the effects of an oral antioxidant-enriched multivitamin supplement in CF disease and clinical outcomes.	Multicenter randomized, double-blind, controlled trial; 73 pancreatic-insufficient subjects with CF 10 years of age and older with an FEV1 between 40% and 100% predicted were randomized to 16 weeks of an antioxidant-enriched multivitamin or control multivitamin without antioxidant enrichment.	Antioxidant supplementation was safe and well tolerated. It increased systemic antioxidant concentrations with a modest reduction in systemic inflammation after 4 weeks. Antioxidant treatment was also associated with a lower risk of first pulmonary exacerbation.
Bergeron et al. 2021 [17]	Review	To summarize the current knowledge of CF genetics and therapies restoring CFTR function, particularly CFTR modulators and gene therapy.	NA	There is hope that the treatment burden can be decreased using highly effective CFTR modulator therapy.
Wu et al. 2003 [21]	Research article	To analyze the role of glutathione in antioxidant defense, nutrient metabolism and regulation of cellular events.	NA	New knowledge on the efficient utilization of dietary protein or precursors for GSH synthesis and its nutritional status is critical for the development of effective therapeutic strategies to treat CF.
Zhao et al. 2019 [22]	Systematic review and meta-analysis	To explore the influence of glutathione versus placebo on pulmonary function in cystic fibrosis.	NA	Glutathione improved pulmonary function in CF, as shown by the increase in FEV1.
Dickerhof et al. 2017 [23]	Original article	To establish whether oxidative stress or glutathione status could be associated with bronchiectasis and whether glutathione deficiency could be linked to CF or a consequence of oxidative stress.	A total of 263 children and infants, out of which 205 had CF and 58 did not. Collectively, they provided 635 BAL samples.	Glutathione deficiency exists in the lower respiratory tract during early stages of cystic fibrosis lung disease, and treatments targeting glutathione have potential benefits for CF patients.
Causer et al. 2020 [24]	Systematic review and meta-analysis.	To evaluate the concentrations of proinflammatory molecules and antioxidant substances in the serum or plasma of CF and non-CF control patients.	Mean contents of blood biomarkers from people with clinically stable CF and non-CF controls were used to calculate the standardized mean difference (SMD) and 95% confidence intervals (95% CI).	Protein carbonyls, F2-isoprostane 8- iso-prostaglandin F2α and malondialdehyde were significantly higher, and vitamins A, β-carotene and albumin were significantly lower in the plasma or serum of people with CF versus controls.
Dickerhof et al. 2020 [25]	Research article	To investigate whether the 2-thioxanthine inhibitor AZM198, when given orally, can inhibit myeloperoxidases in airways of βENaC mice and block oxidative stress without compromising the host’s defense mechanisms.	Transgenic β-epithelial sodium channel (βENaC)-overexpressing mice (n = 10) were infected with *Burkholderia multivorans* and treated twice daily with the MPO inhibitor AZM198.	Blocking hypochlorous acid production in epithelia during pulmonary infections through inhibition of MPO improves morbidity in mice with CF-like lung inflammation without interfering with clearance of bacteria. Inhibition of MPO is an approach to limit oxidative stress in cystic fibrosis lung disease in humans.
Calabrese et al. 2014 [26]	Research article	To evaluate the effect of inhaled GSH in patients with CF.	A total of 54 adult and 51 pediatric patients were randomized to receive inhaled GSH or placebo twice daily for 12 months.	In the pediatric group, a 12-month treatment with inhaled GSH did not lead to any significant increase in FEV1 from baseline. Inhaled GSH has positive effects in CF patients with moderate lung disease.
Hewson et al. 2020 [27]	Research article	To demonstrate that novel antioxidant therapy with the immediate precursor to glutathione, γ-glutamylcysteine (GGC), ameliorates LPS-induced cellular stress in vitro.	Human airway basal epithelial cells were obtained by brushing the nasal inferior turbinate and from bronchoalveolar lavage fluid during bronchoscopy. Proteomic analysis identified perturbations in several pathways related to cellular respiration, transcription, stress responses and cell–cell junction signaling.	Administration of exogenous γ-glutamylcysteine to CF airway epithelium in vitro can increase total intracellular glutathione levels and protect cells from LPS-induced cellular damage.
Pelullo et al. 2020 [28]	Research article	To evaluate if oxidative stress and the aberrant expression levels of genes and microRNAs (miRNAs/miRs) implicated in detoxification may be associated with a better clinical outcome.	Used total RNA extracted from nasal epithelial cells and analyzed the expression levels of oxidative stress genes and one miRNA using quantitative PCR in a representative number of patients with CF compared with healthy individuals.	The activation of an inducible oxidative stress response to protect airway cells against reactive oxygen species injuries in CF patients. The correlations of HO-1 and miR-125b expression with an improved FEV1 value suggested that these factors may synergistically protect airway cells from oxidative stress damage, inflammation and apoptosis.
Di Pietro Caterina et al. 2020 [29]	Review	Blunted heme oxygenase-1 activation in CF-affected cells contributes to hyperinflammation and reduction in the host defense against infections. They discussed potential cellular mechanisms that may lead to decreased heme oxygenase-1 induction in CF cells.	NA	Induction of heme oxygenase-1 may be beneficial for the treatment of CF lung disease. They discussed recent studies highlighting how endogenous heme oxygenase-1 can be induced by administration of controlled doses of CO to reduce lung hyperinflammation, oxidative stress, bacterial infection and dysfunctional ion transport, which are all hallmarks of CF lung disease.
Laselva et al. 2021 [30]	Research article	To understand the role of dimethyl fumarate as an anti-inflammatory and antioxidant drug in CF, they focused on the effect of dimethyl fumarate on CF-related cytokine expression, ROS measurements and CFTR channel function.	Human immortalized bronchial epithelial cells:	Dimethyl fumarate reduced the inflammatory response to LPS stimulation in both CF and non-CF bronchial epithelial cells and restored the LPS-mediated decrease in Kaftrio-TM-mediated CFTR function in CF cells bearing the most common mutation.
Borcherding et al. 2019 [31]	Research article	To determine the effects of CFTR modulation on Nrf2 in primary non-CF and CF human bronchial epithelial cells.	They used primary non-CF or CF human bronchial epithelial cells.	The primary finding of this study is that the F508del CFTR correctors VX809 and VX661 reverse the dysregulation of Nrf2 activity in primary human CF epithelial cells, and that this rescue is CFTR function-dependent.
Nandy Mazumdar et al. 2021 [32]	Research article	Examined the role of BACH1 globally in the oxidative stress response in the airway epithelium and also its role in modulating CFTR expression.	RNA from confluent cultures was extracted with TRIzol (Invitrogen), and cDNA was prepared with the TaqMan reverse transcription kit.	BACH1 regulates CFTR gene expression by modulating locus architecture through its occupancy of enhancers and structural elements, and depletion of BACH1 alters the higher-order chromatin structure. BACH1 may have a dual effect on CFTR expression by direct occupancy of CREs at physiological oxygen (∼8%) while indirectly modulating expression under conditions of oxidative stress.
Scholte et al. 2019 [34]	Research article	To determine whether lipid pathway dysregulation is also observed in BALF from children with CF and to identify biomarkers of early lung disease and potential therapeutic targets.	A comprehensive panel of lipids that included sphingolipids, oxylipins, isoprostanes and lysolipids, all bioactive lipid species known to be involved in inflammation and tissue remodeling, were measured in BALF from children with CF and age-matched non-CF patients with unexplained inflammatory disease	Several lipid biomarkers of early CF lung disease were identified, which point toward potential disease therapeutic approaches used to complement CFTR modulators.
Signorelli et al. 2021 [35]	Research article	To demonstrate that Myriocin modulates the transcriptional profile of CF cells in order to restore autophagy, activate an antioxidative response, stimulate lipid metabolism and reduce lipid peroxidation.	They labeled the cells by means of a fluorescent probe with a high affinity for neutral lipids. We compared CF to healthy cells.	Lipid synthesis is increased in CF, whereas their catabolism is reduced, contributing to inflammation, oxidative stress and impaired autophagy. Myriocin, an inhibitor of sphingolipid synthesis, significantly reduces inflammation. Targeting sphingolipids’ de novo synthesis may counteract lipid accumulation by modulating the CF altered transcriptional profile, thus restoring autophagy and lipid metabolism homeostasis.
Veltman et al. 2021 [36]	Research article	To examine the impact of CFTR deficiency on lipid metabolism and proinflammatory signaling in airway epithelium using a mass spectrometric protein array.	They used CF mouse lung and well-differentiated bronchial epithelial cell cultures of CFTR knockout pigs and CF patients.	Protein array analysis revealed differential expression and shedding of cytokines and growth factors from CF epithelial cells compared to non-CF cells, consistent with sterile inflammation and tissue remodeling under basal conditions.
Olveira et al. 2017 [37]	Research article	To assess oxidation biomarkers and levels of inflammation to determine whether there is an association between these parameters and the intake of macrolides.	Cross-sectional study with clinically stable CF patients and healthy controls. Serum and plasma inflammatory and oxidative stress biomarkers were measured: interleukin-6, reactive C protein, tumor necrosis alpha, glutathione peroxidase, total antioxidant capacity, catalase and superoxide dismutase, together with markers of lipid peroxidation.	Inflammation and oxidation biomarkers were increased in patients with CF compared with controls. The use of azithromycin was associated with reduced TNF-α levels and did not influence oxidation parameters.

## Data Availability

Not applicable.

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
