# Peer review of "Cystic Fibrosis and Oxidative Stress: The Role of CFTR"

_molecules, 2022, doi:10.3390/molecules27165324_

Round 1
Reviewer 1 Report
This well-written paper by Evelina Moliteo, et al., entitled “Cystic fibrosis and oxidative stress: the role of CFTR” which focuses on summarizing the key mechanism by which CFTR deficiency is basically responsible for the pro-inflammatory environment that characterizes what if found in the lung of a patient with CF. The literature search methodology is nicely presented along with the relevant literature related to the oxidative stress in the progression of injury to the CF lung. The summary that is presented in Table 1 is both extensive and exceptionally valuable in providing a remarkable overview of the main studies linking oxidative stress with CF disease. The recognition of current therapeutic approaches, along with the need for the inclusion of a combinatorial drug approaches that are also considerate of local tissue defense mechanism, is excellent.
Author Response
Dear Reviewer, thank you very much for the extremely positive comments that make us very proud of our work.
Reviewer 2 Report
The authors summarized the current literature regarding the oxidative stress related to Cystic Fibrosis. The review lists 37 references comprising original research articles, reviews, and knowledge from databases. The manuscript discusses one of the major features of cystic fibrosis, namely the proinflammatory state of the disease, which is tightly tangled with elevated oxidative conditions with intra- and extracellular redox imbalance. The topic is timely and points out a fundamental translational/clinical challenge in the therapy of the disease.
Critics:
· Lane 35-36: Please clarify this sentence “…..imbalance of both intracellular and extracellular and homeostatic metabolic system.” I think the authors try to compress two pages of knowledge into one sentence. It sounds somewhat mystic than scientific.
· Lane 35: Please change chlorine ion to chloride ion.
· Lane 43-44: Please separate the two mutation classes III and IV. Each class results in a distinct molecular defect and the modern highly effective modulator therapy are not equally effective against mutations in these two groups. It is worth mentioning that mutations belonging to Class III pose abnormal gating and class IV mutation distorts the pore that conducts chloride/bicarbonate ions.
· Lane 53: Please rephrase “lungs suffer the greatest damage”. This statement is incorrect. The GI system with the pancreas is damaged beyond repair. Plugging and inflammation in pancreatic ducts damage the exocrine pancreas function and cause fibrosis (see pancreatic insufficient label among numerous variants in CFTR2 database) and also results in cystic fibrosis-related diabetes in an extremely high proportion of patients. Although these symptoms are manageable and not life-threatening anymore, they are all life-shortening conditions, even with modern supplemental therapy. Also, the vas deferens are entirely missing due to CFTR dysfunction in many patients, leading to infertility in most patients. This damage is not life-threatening, but the damage is beyond repair.
· Lane 132: Please change ATM198 to AZM198.
· Lane 184: Please change m-RNA to mRNA.
· In Table 1:
· at reference Dickerhof et al. in column 3 please change mieloperossidasis to myeloperoxidases and in column 5 epithelial to epithelia.
· At reference Pelullo et al. In column 5 please change patiemts to patients.
· Please add numbering to the references.
Author Response
The authors summarized the current literature regarding the oxidative stress related to Cystic Fibrosis. The review lists 37 references comprising original research articles, reviews, and knowledge from databases. The manuscript discusses one of the major features of cystic fibrosis, namely the proinflammatory state of the disease, which is tightly tangled with elevated oxidative conditions with intra- and extracellular redox imbalance. The topic is timely and points out a fundamental translational/clinical challenge in the therapy of the disease.
Answer: Dear Reviewer, thank you very much for the extremely positive comments that make us very proud of our work. Thanks for the comments and for the review that will allow us to further improve the work.
Critics:
- Lane 35-36: Please clarify this sentence “…imbalance of both intracellular and extracellular and homeostatic metabolic system.” I think the authors try to compress two pages of knowledge into one sentence. It sounds somewhat mystic than scientific.
Answer: Dear Reviewer, as you requested we simplified the sentence.
Lane 35: Please change chlorine ion to chloride ion.
Answer: Thank you, as you requested we corrected the typo.
- Lane 43-44: Please separate the two mutation classes III and IV. Each class results in a distinct molecular defect and the modern highly effective modulator therapy are not equally effective against mutations in these two groups. It is worth mentioning that mutations belonging to Class III pose abnormal gating and class IV mutation distorts the pore that conducts chloride/bicarbonate ions.
Answer: Thank you, we separated the two classes.
- Lane 53: Please rephrase “lungs suffer the greatest damage”. This statement is incorrect. The GI system with the pancreas is damaged beyond repair. Plugging and inflammation in pancreatic ducts damage the exocrine pancreas function and cause fibrosis (see pancreatic insufficient label among numerous variants in CFTR2 database) and also results in cystic fibrosis-related diabetes in an extremely high proportion of patients. Although these symptoms are manageable and not life-threatening anymore, they are all life-shortening conditions, even with modern supplemental therapy. Also, the vas deferens are entirely missing due to CFTR dysfunction in many patients, leading to infertility in most patients. This damage is not life-threatening, but the damage is beyond repair.
Answer: Dear Reviewer, yes, it's true. Your reasoning is acceptable. For this reason, we have changed the sentence.
Lane 132: Please change ATM198 to AZM198.
Answer: Thank you, as you requested we corrected the typo.
- Lane 184: Please change m-RNA to mRNA.
Answer: Thank you, as you requested we corrected the typo.
In Table 1:
- at reference Dickerhof et al. in column 3 please change mieloperossidasis to myeloperoxidases and in column 5 epithelial to epithelia.
Answer: Thank you, as you requested we corrected the typo.
- At reference Pelullo et al. In column 5 please change patiemts to patients.
Answer: Thank you, as you requested we corrected the typo.
- Please add numbering to the references.
Answer: Thank you. We added the numbers of references.